# Smoking Cessation Rate and Its Predictors among Heavy Smokers in a Smoking-Free Hospital in Taiwan

**DOI:** 10.3390/ijerph182412938

**Published:** 2021-12-08

**Authors:** Chin-Jung Lin, Wei-Hsin Huang, Che-Yuan Hsu, Jin-Jin Tjung, Hsin-Lung Chan

**Affiliations:** 1Department of Family Medicine, Mackay Memorial Hospital, Taipei 106, Taiwan; junglin20022002@yahoo.com.tw (C.-J.L.); whh5881@gmail.com (W.-H.H.); passionhsu1120@gmail.com (C.-Y.H.); jinjin0921@gmail.com (J.-J.T.); 2Community Health Center, Mackay Memorial Hospital, Taipei 106, Taiwan; 3Division of Medicine, Mackay Medical College, New Taipei City 252, Taiwan

**Keywords:** heavy smoker, nicotine replacement therapy, quit smoking clinic, smoking cessation, varenicline, predictor

## Abstract

Smoking poses critical risks for heart disease and cancers. Heavy smokers, defined as smoking more than 30 pack-year, are the most important target for smoking cessation. This study aimed to obtain the cessation rate and its predictors among heavy smokers. We collected data from heavy smokers who visited a smoking-free hospital in Taiwan during 2017. All patients were prescribed either varenicline or nicotine replacement therapy (NRT) for smoking cessation, and their smoking status was followed for six months. Successful smoking cessation was defined by self-reported no smoking over the preceding seven days (7-day point abstinence). In total, 280 participants with a mean aged of 53.5 years were enrolled, and 42.9% of participants successfully stopped smoking in 6 months. The results revealed that quitters were older, with hypertension, fewer daily cigarettes, and being prescribed with varenicline. Multiple logistic regressions analyses identified that fewer daily cigarettes and being prescribed with varenicline were predictors of successful smoking cessation. Therefore, we suggest that varenicline use may help heavy smokers in smoking cessation.

## 1. Introduction 

Tobacco use is one of the leading public health problems that causes more than 8 million deaths per year in the world. Based on the World Health Organization statistics, 7 million deaths per year are related to direct tobacco use, and about 1 million deaths are related to exposure to secondhand smoke [1]. According to the National Health Interview Survey 2017 data, about 47.4 million adults are current smokers in the United States [2]. The prevalence of smoking in Taiwan is approximately 3.5 million, and more than 20,000 people died every year in ways related to tobacco use [3]. The rate of death from any cause among current smokers was about three times that among those who had never smoked, and smokers lose at least one decade of life expectancy compared with those who have never smoked [4].

Smoking intensity consists of smoking amount and smoking duration. It was calculated by multiplying the number of packs of cigarettes smoked per day by the number of years the subject has smoked. Thirty pack-year means smoking one pack per day for 30 years or two packs per day for 15 years. In the prospective cohort study, the daily cigarette count was associated with sudden cardiac death even without coronary heart disease at baseline [5]. In addition, smoking intensity posed a positive dose-response relationship with significant increases in lymphoma risk [6]. Based on the dose-response meta-analysis, the risk of stroke increased by 12% for each increment of five cigarettes per day [7]. In non-small cell lung cancer, the increase in pack-years was associated with decreased median overall survival [8]. In 2013, the American Cancer Society recommended lung cancer screening with low-dose computer tomography in persons aged 55 to 74 years who have ≥a 30 pack-year smoking history, which was defined as a high-risk population [9]. 

Heavy smokers seem to be the most important target for smoking cessation, but few studies evaluate the predictors of smoking cessation in this group. In this study, we defined heavy smokers as smoking intensity ≥ 30 pack-year. The purpose of the study was to find out the smoking cessation rate and its predictors of successful smoking cessation among heavy smokers in our smoking-free hospital.

## 2. Materials and Methods

### 2.1. Study Populations/Subjects

We performed a retrospective study from records of patients visiting the outpatient smoking cessation clinics in Mackay Memory Hospital in Taiwan from 1 January to 31 December 2017. All participants were over the age of 18 years old and were legally covered in the National Health Insurance Program in Taiwan, had a Fagerström Test of Nicotine Dependence (FTND) score equal to or greater than 4 or smoked more than 10 cigarettes per day. A total of 280 heavy smokers (smoking history ≥ 30 pack-years) were included in the study. All participants were prescribed either varenicline or nicotine replacement therapy (NRT) for smoking cessation. Counseling for smoking cessation were provided by physicians and case managers. Telephone interviews of individual participants were conducted by case managers at the end of 2 weeks, 3 months, and 6 months after the participants’ first clinic visit to assess their smoking condition and drug adherence. 

### 2.2. Data Collection and Outcome Measures

The basic demographic data, including age, gender, body weight, medical history, and smoking history were obtained at the first outpatient clinic visit. Participants were asked about their smoking habits, including daily consumption of cigarettes and how many years have they been smoking. The FTND was used to assess their smoking dependence (scores range from 0 to 10, with higher scores representing greater nicotine dependence). In general, smokers with FTND ≥ 7 may have strong withdrawal symptoms and relapse early [10]. Exhaled CO levels were assessed with a carbon monoxide monitor.

The participants were followed with telephone calls by case managers to collect information about their smoking status six months after the treatment. Those who self-reported as having stopped smoking were assumed to have successfully stopped smoking. Successful smoking cessation was defined by self-reported no smoking over the preceding seven days (7-day point abstinence) at six months follow-up.

### 2.3. Statistical Analysis

Data are expressed as means ± SDs for continuous variables, and as percentages for categorical variables. We performed Student’s *t*-test and a Chi-square test to detect significant differences among the variables between quitters and non-quitters. Multivariate logistic regression analysis was applied for assessments of the odds ratio (OR) of associated factors for smoking cessation. All statistical analyses were conducted using SPSS version 22.0(IBM Corp., Armonk, NY, USA). The criterion for statistical significance was *p* < 0.05.

### 2.4. Ethical Approval

The study was performed with the approval of the Institutional Review Board of Mackay Memorial Hospital, Taipei, Taiwan (application number, 17MMHIS049).

## 3. Results

The characteristics of demographic data are shown in Table 1. The participants were mainly male (82.5%), with a mean age of 53.5 years old. Because the participants were heavy smokers, the mean smoking years was 33, the average daily cigarettes count was 30.4, and the FTND score was 7.4. The average pharmacotherapy treatment duration was 5.9 ± 4.3 weeks. In all, 219 (78.2%) participants received varenicline, and 61 (21.8%) participants received NRT. The highest proportion of comorbidity was hyperlipidemia (36.4%) and followed by hypertension (33.9%).

The comparison of baseline characteristics among quitters and non-quitters is shown in Table 2. Overall, there were 120 (42.9%) participants who had successfully quit smoking within six months. Compared with non-quitters, quitters were older, smoking less than 30 cigarettes per day, prescribed with varenicline, and with comorbidity of hypertension. There was no significant difference between the quitters and non-quitters in terms of gender, body weight, smoking duration, exhaled CO concentration, FTND, duration of pharmacotherapy treatment, comorbidity of diabetes, hyperlipidemia, heart disease, lung disease, neurological disease, and cancer.

With multiple logistic regression analysis for odds ratio (OR) of associated factors for smoking cessation, the results showed heavy smokers who used varenicline and successfully quit smoking in 6 months outnumbered those who used NRT (OR: 2.44, 95% Confidence interval (CI): 1.26–4.74) and the daily cigarette count was negatively associated with smoking cessation (OR: 0.97, 95% CI: 0.95–0.99) (Table 3).

## 4. Discussion

Studies have emphasized a strong dose-dependent relationship between smoking amount and related diseases’ outcomes [7,11]. Heavy smokers have higher risks for associated diseases, so the physicians should advise them more actively to quit smoking. According to a previous study, heavy smokers have some characteristics, such as being male, have used quitting aids before, have other substances used, more alcohol abuse, depression, and lower confidence in quitting [12]. Without being advised by health professionals, the heavy smokers have a relative lower smoking cessation rate. Smokers who undergo smoking cessation programs, a combination of drug therapies and behavioral change instruction are five times more successful in quitting smoking than smokers who attempt to quit on their own [13]. In 2017, an average 6-month success rate of stopping smoking is 28.8% according to the Health Promotion Administration in Taiwan [14]. In our study, 42.9% of heavy smokers had successfully stopped smoking within 6 months. The higher success rate may be related to the explanation concerning the benefits of stopping smoking, the doctor’s encouragement, the case manager’s regular follow-up telephone call, the lesser adverse reaction of varenicline, and the payment of only USD 10 per visit. Although the smoking cessation rate of heavy smokers was low, active promotion of smoking cessation programs can still help increase the cessation rate [15].

One study showed that smoking cessation was more common with increasing age, higher education and fewer years of smoking. Asthma, wheeze, hay fever, chronic bronchitis, diabetes, and hypertension did not significantly predict smoking cessation, but smokers hospitalized for ischemic heart disease during the study period were more prone to stopping smoking [16]. In our study, older age and comorbidity of hypertension were significantly different between quitters and non-quitters, but the logistic regression analysis did not show that they were the predictors of smoking cessation. Nicotine in cigarettes may increase the blood pressure. Quitting smoking and lowering blood pressure are more likely to be noticed by smokers than lowering blood sugar or lipids. This might be the reason why comorbidity of hypertension was significantly different between quitters and non-quitters. 

One study in Taiwan stratified FTND (low: 0–3; medium: 4–6; high: ≥7) and indicated a lower FTND is a predictor of success smoking cessation [17]. However, our study showed the average FTND score among the heavy smokers was 7.4 and there was no significant difference between quitters and non-quitters based on FTND. The reason might be that the participants were all heavy smokers with high FTND level. The reason for higher smoking cessation rate in men is that women have more concerns about the negative outcome of smoking cessation, including weight gain [18]; however, in our study, gender had no significant difference in smoking cessation among heavy smokers

An analysis of cohort study, lower levels of daily cigarettes count was the predictor of smoking cessation [19]. In our study, the mean daily cigarettes count among heavy smokes was 30.4 (SD 11.8). There was significant difference between quitters and non-quitters in terms of daily cigarettes fewer than or more than 30 per day. Logistic regression analysis also showed fewer daily cigarettes was the predictor of smoking cessation (OR: 0.97 [0.95–0.99]).

In clinical settings, smoking cessation medication was prescribed individually according to each participant’s medical condition. In our study, 120 (42.9%) participants successfully quit smoking in 6 months and the proportion of quitters using varenicline was significantly higher than that of non-quitters. The study revealed that varenicline use might help in smoking cessation in heavy smokers compared to NRT. Previous studies also show that comparison of varenicline with NRT or bupropion, varenicline has significantly greater efficacy for smoking cessation [20,21,22]. Varenicline reduces the symptoms of nicotine withdrawal by binding with high affinity and acting as a partial agonist at the alpha-4 beta-2 nicotine receptor. Through its stimulating effects of the receptor, it reduces the cravings of smokers. Varenicline also blocks nicotine from binding to the receptor, interrupting the reinforcing effects of nicotine that lead to nicotine dependence. Through this action, it reduces the rewarding aspects of cigarettes smoking. In our study, the average FTND score of participants was 7.4, which means a high dependence and craving for nicotine. Varenicline’s mechanism of action might be the reason why varenicline is better than NRT for heavy smokers. 

## 5. Conclusions

Heavy smokers pose a higher risk of associated diseases, so the physicians should put more effort into these populations on smoking cessation. Heavy smokers with fewer daily cigarettes were more likely to quit smoking. We also suggest that varenicline use may help in smoking cessation among heavy smokers. 

## Figures and Tables

**Table 1 ijerph-18-12938-t001:** Participants’ baseline characteristics.

Variable	Total *n* = 280
	Mean ± SD or N(%)
Demographics	
Age (years)	53.5 ± 10.1
Men	231 (82.5%)
Weight (kg)	71.6 ± 14.9
Smoking years (years)	33.0 ± 9.9
Cigarettes/day	30.4 ± 11.8
Exhaled CO (ppm)	19.6 ± 11.9
FTND score	7.4 ± 2.3
Pharmacotherapy duration (weeks)	5.9 ± 4.3
Pharmacotherapy with Varenicline	219 (78.2%)
Pharmacotherapy with NRT	61 (21.8%)
Comorbidities	
Hypertension	95 (33.9%)
Diabetes	63 (22.5%)
Hyperlipidemia	102 (36.4%)
Heart disease	39 (13.9%)
Lung disease	15 (5.4%)
Neurological disease	19 (6.8%)
Cancer	15 (6.4%)

Abbreviations: CO (ppm) = carbon monoxide (parts per million); FTND score = Fagerström Test of Nicotine Dependence score; NRT = nicotine replacement therapy.

**Table 2 ijerph-18-12938-t002:** Comparison of baseline characteristics related to smoking cessation among quitters and non-quitters.

	Quitters (*n* = 120) (42.9%)	Non-Quitters (*n* = 160) (57.1%)	*p* Value *
Variable	Mean ± SD or N(%)		
Demographics			
Age (years)	55.4 ± 9.9	50.5 ± 10.1	0.0058
Men	100 (83.3%)	131 (81.9%)	0.7506
Weight (kg)	71.9 ± 14.7	71.4 ± 15.1	0.8104
Smoking years (years)	34.0 ± 10.0	32.3 ± 9.8	0.1392
Cigarettes/day	27.9 ± 10.5	32.2 ± 12.5	0.0025
Exhaled CO (ppm)	18.1 ± 12.3	20.8 ± 11.5	0.0525
FTND score	7.2 ± 2.3	7.5 ± 2.3	0.1872
Pharmacotherapy duration (weeks)	6.3 ± 4.4	5.5 ± 4.3	0.132
Pharmacotherapy with Varenicline	104 (86.7%)	115 (71.9%)	0.003
Pharmacotherapy with NRT	16 (13.3%)	45 (28.1%)	0.114
Comorbidities			
Hypertension	49 (40.8%)	46 (28.8%)	0.0346
Diabetes mellitus	24 (20.0%)	39 (24.4%)	0.3856
Hyperlipidemia	48 (40.0%)	54 (33.8%)	0.2822
Heart disease	18 (15.0%)	21 (13.1%)	0.6538
Lung disease	8 (6.7%)	7 (4.4%)	0.3994
Neurologic disease	9 (7.5%)	10 (6.3%)	0.6807
Cancer	7 (5.8%)	8 (5.0%)	0.7593

Abbreviations: CO (ppm) = carbon monoxide (parts per million); FTND score = Fagerström Test of Nicotine Dependence score; NRT = nicotine replacement therapy; * Using Chi-square and *t*-test.

**Table 3 ijerph-18-12938-t003:** Odds ratio (95% confidence interval) among quitters and non-quitters’ groups with logistic regression analysis.

	Quitters vs. Non-Quitters	
	Odds Ratio (95% CI)	*p* Value
Age (years)	1.02 (0.99–1.05)	0.171
Hypertension	1.42 (0.82–2.46)	0.205
Daily cigarette count	0.97 (0.95–0.99)	0.041
Pharmacotherapy (Varenicline vs. NRT)	2.44 (1.26–4.74)	0.008

Abbreviations: FTND score = Fagerström Test of Nicotine Dependence score; NRT = nicotine replacement therapy.

## Data Availability

A retrospective study from records of patients visiting the outpatient smoking cessation clinics in Mackay Memory Hospital in Taiwan from 1 January to 31 December 2017.

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
