# Peer review of "Smoking Cessation Rate and Its Predictors among Heavy Smokers in a Smoking-Free Hospital in Taiwan"

_ijerph, 2021, doi:10.3390/ijerph182412938_

Round 1

Reviewer 1 Report

It describes a sample of smokers with % success in quitting smoking, but the rest I think is of little interest: It is not clear how the treatments were chosen for the patients; there may be a significant bias because there is no randomisation for this, and therefore no conclusions can be drawn about the best treatment.
Most of the results are as expected and already known, and others are poorly explained: how is it justified that the fagerstrom test has no influence on the success rate? The discussion could be improved.

Author Response

Reviewer 1

Comments and Suggestions for Authors

It describes a sample of smokers with % success in quitting smoking, but the rest I think is of little interest: It is not clear how the treatments were chosen for the patients; there may be a significant bias because there is no randomisation for this, and therefore no conclusions can be drawn about the best treatment.

Most of the results are as expected and already known, and others are poorly explained: how is it justified that the fagerstrom test has no influence on the success rate? The discussion could be improved.

Ans: Thank you for your comments. The key point of this study is that we found the use of varenicline in heavy smokers may increase the cessation success rate. In clinical settings, smoking cessation medication was prescribed individually according to each participant’s medical condition. Therefore, in smoking cessation clinics, we do not randomize patients’ treatment. This is a real world study and the conclusion of this study can be drawn is that we suggest the clinicians to use varenicline, instead of NRT, for the treatment of smoking cessation among heavy smokers which was a group less discussed before. Because we focused on heavy smokers, the mean Fagerstrom score was 7.4 and there is no statistical difference between quitters and non-quitters. We found that Fagerstrom test has no influence on the success rate in heavy smokers. We also revised and improved the discussion.

Reviewer 2 Report

The authors studied 280 smokers admitted to a smoke free hospital in 2017 with >= 30 pack year smoking histories, FTND>5, national health insurance, and without several acute illness exclusion criteria. Smokers were identified on admission then referred to a smoking cessation clinic where they were evaluated and treated with 1-2 rounds of behavior therapy and either varenicline or NRT. After 6 months 120 of these smokers reported 7-day abstinence. A logistic regression identified older age, lower daily cigarette consumption, varenicline treatment, and the presence of hypertension as significant predictors of smoking cessation.

Major Issues

This is not a controlled trial. As an observational study of smoking cessation in a developed country at this point in history, the authors need to highlight some new insight or discrepancy that will prompt further investigation, or remind us of some important idea that has been forgotten or ignored. For instance, what easily identifiable subgroups are especially likely to quit? We can focus available therapies on them. What subgroups are especially unlikely to quit? We need to develop new strategies for them, and not waste resources repeating treatments that do not help.

The report does not thoroughly describe participant recruitment. 

How many smokers were admitted to the smoke free hospital?

How many met each of the inclusion criteria (insurance, pack-years, FTND)?

How many met all three inclusion criteria?

How many of these were excluded for an acute or otherwise disqualifying illness?

How many were referred to the clinic?

How many made at least 1 clinic visit?

How many finished 1 round of behavioral therapy?

How many began a second round of behavioral therapy?

I expect that there are substantial declines in eligibility or participation at several of these steps. This will likely lead to some reflection regarding the extreme challenges people face when they try to quit smoking. A 42% cessation rate is pretty high, even for self report and varenicline at 6 months, but the fraction of smokers or heavy smokers who actually quit will be smaller.

The study probably has too few observations to support confident inferences regarding women, stable psychiatric illness, and NRT, especially as a treatment for clinically identifiable subgroups. For instance, the p value on stable psychiatric illness affecting the odds of cessation in this study is 0.057, but based on other work the prior probability that psychiatric illness affects smoking cessation is quite high.  This is probably just a statistical power problem, with 62 persons having stable psychiatric illness. 

Other issues

In the first paragraph of the introduction, the prevalence of smoking in the USA is not nearly as interesting or relevant as the prevalence in mainland China, which I believe has more smokers than the USA has people. Furthermore, Taiwan’s population is genetically and culturally more similar to mainland China than to the USA.

Were any hospital patients using electronic cigarettes? If so, how were these people handled?

If at all possible, the authors might want to consider genotype data, or phenotype data that might reflect genotype. Smoking genetics is widely ignored in cessation studies, but it may be possible for even observational studies to inform discussions about optimal treatments for various combinations of addiction susceptibility and nicotine metabolism rates. 

There is a lot of overlap between individuals smoking 28+/-10 versus 32+/-12 cigarettes per day. Again, some sort of subgroup analysis might help us think about who can benefit from different interventions. 

Author Response

Reviewer 2

*This is not a controlled trial. As an observational study of smoking cessation in a developed country at this point in history, the authors need to highlight some new insight or discrepancy that will prompt further investigation, or remind us of some important idea that has been forgotten or ignored. For instance, what easily identifiable subgroups are especially likely to quit? We can focus available therapies on them. What subgroups are especially unlikely to quit? We need to develop new strategies for them, and not waste resources repeating treatments that do not help.

Ans: Thank you for your comments. This is and retrospective observational study about smoking cessation among heavy smokers. The highlight of this study is that we found the subgroup who smoke fewer daily cigarettes are more likely to quit, and the clinicians may use varenicline for the treatment of smoking cessation.  

*The report does not thoroughly describe participant recruitment.

Ans: We revised the description of study population/subjects.

“We performed a retrospective study from records of patients visiting the outpatient smoking cessation clinics in Mackay Memory Hospital in Taiwan from 1 January to 31 December 2017. All participants were over the age of 18 years old and were legally covered in the National Health Insurance Program in Taiwan, with Fagerström Test of Nicotine Dependence (FTND) score equal to or greater than 4 or smoked more than 10 cigarettes per day. A total of 280 heavy smokers (smoking history ≥ 30 pack-years) were included in the study.”

*How many smokers were admitted to the smoke free hospital?

Ans: The study population were from outpatients smoking cessation clinics.

*How many met each of the inclusion criteria (insurance, pack-years, FTND)?

*How many met all three inclusion criteria?

Ans: All participants visiting outpatient smoking cessation clinics were covered in the National Health Insurance Program in Taiwan, with FTND equal to or greater than 4 or smoked more than 10 cigarettes per day, and 280 heavy smokers (smoking history ≥ 30 pack-years) were included in the study.

*How many of these were excluded for an acute or otherwise disqualifying illness?

Ans: All participants were from outpatient smoking cessation clinics. No participants were excluded for acute or otherwise disqualifying illness.

*How many were referred to the clinic?

Ans: All participants were from outpatient smoking cessation clinics.

*How many made at least 1 clinic visit?

Ans: All participants had at least 1 clinic visit.

*How many finished 1 round of behavioral therapy?

*How many began a second round of behavioral therapy?

Ans: We revised the manuscript and used “counseling” instead of “behavioral therapy”. All participants received counseling for smoking cessation provided by physicians and case managers.

*I expect that there are substantial declines in eligibility or participation at several of these steps. This will likely lead to some reflection regarding the extreme challenges people face when they try to quit smoking. A 42% cessation rate is pretty high, even for self report and varenicline at 6 months, but the fraction of smokers or heavy smokers who actually quit will be smaller.

Ans: Thank you for your comment. Since 2002, the Health Promotion Administration (HPA) in Taiwan had provided subsidies for smoking cessation services. Smokers above 18 years of age with scoring at least four points on FTND or smoking 10 or more cigarettes per day may undergo two treatment courses each year. Each treatment course provides up to eight weeks of medication, counseling, and subsidies for each clinic visit. Smokers are required to pay a maximum co-payment of only NT$ 300 (US$ 10) for each visit. Therefore, lots of smokers in Taiwan visited smoking cessation clinics and we collected plenty of information from participants.

*The study probably has too few observations to support confident inferences regarding women, stable psychiatric illness, and NRT, especially as a treatment for clinically identifiable subgroups. For instance, the p value on stable psychiatric illness affecting the odds of cessation in this study is 0.057, but based on other work the prior probability that psychiatric illness affects smoking cessation is quite high.  This is probably just a statistical power problem, with 62 persons having stable psychiatric illness.

Ans: Thank your for your comment. After consideration, we decided to delete the variable “stable psychiatric illness”. The definition of stable psychiatric illness is too rough and not clear. Moreover, the diagnosis of stable psychiatric illness is not as simple as other comorbidities, such as hypertension, diabetes, hyperlipidemia and cancer.  

Other issues

*In the first paragraph of the introduction, the prevalence of smoking in the USA is not nearly as interesting or relevant as the prevalence in mainland China, which I believe has more smokers than the USA has people. Furthermore, Taiwan’s population is genetically and culturally more similar to mainland China than to the USA.

Ans: Thank you for your comment. According to WHO report, the prevalence of smoking in USA was 14% and 23% in China in 2019. According to the report of Health Promotion Administration in Taiwan, the prevalence of smoking in 2020 was 13.1%. The prevalence of smoking in Taiwan is much lower than the prevalence in China and close to USA. Though Taiwan’s population is genetically and culturally more similar to mainland China, we have put a lot of effort in tobacco control in Taiwan and attained a great achievement.

*Were any hospital patients using electronic cigarettes? If so, how were these people handled?

Ans: According to the report of Health Promotion Administration in Taiwan, the prevalence of electronic cigarettes in 2020 was 1.7%. The electronic cigarette or vaping product use associated lung injury (EVALI) was first reported by CDC in 2019. We did not survey how many patients use e-cigarettes in 2017, but we already have started to collect the information of the use of e-cigarettes in patients in our hospital.

*If at all possible, the authors might want to consider genotype data, or phenotype data that might reflect genotype. Smoking genetics is widely ignored in cessation studies, but it may be possible for even observational studies to inform discussions about optimal treatments for various combinations of addiction susceptibility and nicotine metabolism rates.

Ans: Thank you for your suggestion. Indeed, there was a study discussing about the genetic nicotine metabolism rates performed in Taiwan in our hospital and other cooperated hospitals several years ago. However, the study failed because of few participants. Smokers usually ask for treatment and counseling for smoking cessation directly, they might not willing being drawn blood for genetic test. That’s why there are few studies about genotype data in smoking cessation.

*There is a lot of overlap between individuals smoking 28+/-10 versus 32+/-12 cigarettes per day. Again, some sort of subgroup analysis might help us think about who can benefit from different interventions.

Ans: Thank you for your comment. We found fewer daily cigarettes were more likely to quit smoking in heavy smokers. As you mentioned, the overlap might be the reason of borderline significant difference in logistic regression analysis.

Reviewer 3 Report

This paper documents results from a smoking cessation program aimed specifically at heavy smokers (individuals who have been smoking an average of 30 cigarettes a day for ~30 years). The cessation program included behavioral counselling and pharmaceutical treatment with either nicotine replacement therapy (patch) OR varenicline for 8 weeks. The study concludes that varenicline is better than nicotine replacement therapy for helping heavy smokers quit smoking.  Overall, this is well-designed study where conclusions are supported by data. The manuscript needs significant editing support for English language.

Major concerns:

According to CDC, as the smoking rates are going down, vaping is increasing. Thus it is relevant to ask the following questions: How common was the e-cigarette use in study participants? How many were vaping before enrollment in the study (dual smokers) and how many took up e-cigarettes as a smoking cessation aid? What flavors of e-cigarettes were popular? Did e-cigarettes contain nicotine or not?

Hypertension is mentioned as a predictor of smoking cessation. Is this new diagnosis of hypertension at the time of inclusion in the study? If the subjects knew about their hypertension diagnosis prior to enrollment in this study, why did it affect their decision to quit smoking during study duration? Was the hypertension, diabetes, etc discussed during counselling? Or is hypertension just a confounding predictor detected in this study?

Minor concerns:

Cite references for the statement, "The history of more cigarette smoking was related to disease survival rate."

Extensive English language editing is recommended. Some (NOT ALL) examples are below:

1. 8 million people deaths should be replaced with 8 million deaths

2. According to the New England Journal of Medicine 2013 with ten thousand data,

3. Heavy smokers seem to be the most important target for smoking cessation, but few 53 studies evaluate the predictors of smoking cessation in this group.

4. current smokes patients were suggested and transfer to the quit smoking clinic

What do authors mean by “ statistically significant upward curvature”?

What is smoking year? –Line 76

The following statement is difficult to understand. Please edit: After consultation with the patients concerning the use of the drugs, and by excluding the use for those with previous drug allergies, the use of varenicline for those with suicidal tendencies or with a history of seizures, the use of nicotine gum for patients with dental problems or temporomandibular joint disease, and the use of nicotine patches for patients with skin allergies or dermatologic conditions, physicians prescribed appropriate drugs for a treatment course lasting a maximum of eight weeks.

Author Response

Reviewer 3

Open Review

(x) I would not like to sign my review report

( ) I would like to sign my review report

English language and style

(x) Extensive editing of English language and style required

( ) Moderate English changes required

( ) English language and style are fine/minor spell check required

( ) I don't feel qualified to judge about the English language and style

Yes   Can be improved     Must be improved   Not applicable

Does the introduction provide sufficient background and include all relevant references?

( )     (x)    ( )     ( )

Is the research design appropriate?

(x)    ( )     ( )     ( )

Are the methods adequately described?

(x)    ( )     ( )     ( )

Are the results clearly presented?

(x)    ( )     ( )     ( )

Are the conclusions supported by the results?

(x)    ( )     ( )     ( )

Comments and Suggestions for Authors

This paper documents results from a smoking cessation program aimed specifically at heavy smokers (individuals who have been smoking an average of 30 cigarettes a day for ~30 years). The cessation program included behavioral counselling and pharmaceutical treatment with either nicotine replacement therapy (patch) OR varenicline for 8 weeks. The study concludes that varenicline is better than nicotine replacement therapy for helping heavy smokers quit smoking.  Overall, this is well-designed study where conclusions are supported by data. The manuscript needs significant editing support for English language.

Major concerns:

*According to CDC, as the smoking rates are going down, vaping is increasing. Thus it is relevant to ask the following questions: How common was the e-cigarette use in study participants? How many were vaping before enrollment in the study (dual smokers) and how many took up e-cigarettes as a smoking cessation aid? What flavors of e-cigarettes were popular? Did e-cigarettes contain nicotine or not?

Ans: Thank you for your comment and suggestion. According to the report of Health Promotion Administration in Taiwan, the prevalence of electronic cigarettes in 2020 was 1.7%. The electronic cigarette or vaping product use associated lung injury (EVALI) was first reported by CDC in 2019. We did not survey how many patients use e-cigarettes in 2017, but we already have started to collect the information of the use of e-cigarettes in smokers in our hospital.

*Hypertension is mentioned as a predictor of smoking cessation. Is this new diagnosis of hypertension at the time of inclusion in the study? If the subjects knew about their hypertension diagnosis prior to enrollment in this study, why did it affect their decision to quit smoking during study duration? Was the hypertension, diabetes, etc discussed during counselling? Or is hypertension just a confounding predictor detected in this study?

Ans: Thank you for your comment. Hypertension is a comorbidity prior to enrollment in this study. Physicians discussed with smokers about the tobacco effect of hypertension, diabetes, etc during counselling. We put our assumption in the discussion: “Nicotine in cigarettes may increase the blood pressure. Quitting smoking and lowering blood pressure are more likely to be noticed by smokers than lowering blood sugar or lipid. This might be the reason why comorbidity of hypertension was significant different between quitters and non-quitters.”

Minor concerns:

Cite references for the statement, "The history of more cigarette smoking was related to disease survival rate."

à We deleted this sentence.

Extensive English language editing is recommended. Some (NOT ALL) examples are below:

  1. 8 million people deaths should be replaced with 8 million deaths

Ans: Thank you, we corrected this sentence.

  1. According to the New England Journal of Medicine 2013 with ten thousand data,

Ans: Thank you, we revised the statement.

  1. Heavy smokers seem to be the most important target for smoking cessation, but few 53 studies evaluate the predictors of smoking cessation in this group.

Ans: Thank you, the “53” might be the line number.

  1. current smokes patients were suggested and transfer to the quit smoking clinic

Ans: Thank you, we revised the statement and deleted this sentence.

What do authors mean by “ statistically significant upward curvature”?

Ans: Thank you, we changed the reference and revised the statement.

What is smoking year? –Line 76

Ans: We revised the sentence: How many years have they been smoking?

*The following statement is difficult to understand. Please edit: After consultation with the patients concerning the use of the drugs, and by excluding the use for those with previous drug allergies, the use of varenicline for those with suicidal tendencies or with a history of seizures, the use of nicotine gum for patients with dental problems or temporomandibular joint disease, and the use of nicotine patches for patients with skin allergies or dermatologic conditions, physicians prescribed appropriate drugs for a treatment course lasting a maximum of eight weeks.

Ans: Thank you for your comment. The statements are all deleted. We revised the statements.

Round 2

Reviewer 1 Report

I don't think we can conclude from this study that we should use varenicline instead of NRT in heavy smokers. 

Author Response

Reviewer 1

Comments and Suggestions for Authors

I don't think we can conclude from this study that we should use varenicline instead of NRT in heavy smokers. 

Ans: Thank you for your comment. We add our hypothesis and description at the end of the discussion: “Varenicline reduces the symptoms of nicotine withdrawal by binding with high affinity and acting as a partial agonist at the alpha-4 beta-2 nicotine receptor. Through its stimulating effects of the receptor, it reduces the cravings of smokers. Varenicline also blocks nicotine from binding to the receptor, interrupting the reinforcing effects of nicotine that lead to nicotine dependence. Through this action, it reduces the rewarding aspects of cigarettes smoking. In our study, the average FTND score of participants was 7.4, which means a high dependence and craving for nicotine. Varenicline’s mechanism of action might be the reason why varenicline is better than NRT for heavy smokers.”  We did not emphasize that doctors should use varenicline instead of NRT in heavy smokers. We only suggest that varenicline use may help heavy smokers in smoking cessation.